# Understanding Patterns of the Gut Microbiome May Contribute to the Early Detection and Prevention of Type 2 Diabetes Mellitus: A Systematic Review

**DOI:** 10.3390/microorganisms13010134

**Published:** 2025-01-10

**Authors:** Natalia G. Bednarska, Asta Kristine Håberg

**Affiliations:** 1London School of Hygiene and Tropical Medicine, London WC1E 7HT, UK; 2Department Neuromed & Movement Science, Norwegian University of Science & Technology (NTNU), 7034 Trondheim, Norway; asta.haberg@ntnu.no

**Keywords:** gut microbiome, diabetes, hyperglycaemia, prediabetes, microflora, type 2 diabetes mellitus, gut flora, kynurenine, *Akkermansia muciniphila*, metformin, GLP-1

## Abstract

The rising burden of type 2 diabetes mellitus (T2DM) is a growing global public health problem, particularly prominent in developing countries. The early detection of T2DM and prediabetes is vital for reversing the outcome of disease, allowing early intervention. In the past decade, various microbiome–metabolome studies have attempted to address the question of whether there are any common microbial patterns that indicate either prediabetic or diabetic gut microbial signatures. Because current studies have a high methodological heterogeneity and risk of bias, we have selected studies that adhered to similar design and methodology. We performed a systematic review to assess if there were any common changes in microbiome belonging to diabetic, prediabetic and healthy individuals. The cross-sectional studies presented here collectively covered a population of 65,754 people, with 1800 in the 2TD group, 2770 in the prediabetic group and 61,184 in the control group. The overall microbial diversity scores were lower in the T2D and prediabetes cohorts in 86% of the analyzed studies. Re-programming of the microbiome is potentially one of the safest and long-lasting ways to eliminate diabetes in its early stages. The differences in the abundance of certain microbial species could serve as an early warning for a dysbiotic gut environment and could be easily modified before the onset of disease by changes in lifestyle, taking probiotics, introducing diet modifications or stimulating the vagal nerve. This review shows how metagenomic studies have and will continue to identify novel therapeutic targets (probiotics, prebiotics or targets for elimination from flora). This work clearly shows that gut microbiome intervention studies, if performed according to standard operating protocols using a predefined analytic framework (e.g., STORMS), could be combined with other similar studies, allowing broader conclusions from collating all global cohort studies efforts and eliminating the effect-size statistical insufficiency of a single study.

## 1. Introduction

It has been estimated that, globally, the prevalence of T2D mellitus will increase to 7079 individuals per 100,000 by 2030 [1]. In the USA, around 1 in 14 adults are estimated to have prediabetes, whilst in the UK, 1 in 9 adults are affected, often without being aware of it [2,3]. Current preventative and treatment strategies for prediabetes and diabetes are limited and do not always bring the desired results [4]. Since diabetes starts with chronic mild hyperglycaemia, so called prediabetes (defined as fasting plasma glucose of 6.1–7.0 mmol/L or HbA_1c_ of 42–48 mmol/mol [6.0–6.5%]) [5], it is beneficial to capture this early phase. Changes in microbiome, lifestyle and diet are the least invasive ways to stop glucose desensitization and to cease the further onset of diabetes.

The gut microbiome consists of communities of metabolically active microorganisms, the growth of which is either promoted or limited by the host’s diet (including prebiotics), lifestyle and health factors. Each person’s microbial patterns differ in abundance and composition, therefore giving them a unique fingerprint [6]. However, during recent years of microbiome field research, common patterns of microbes have been distinguished and associated with eubiosis (which reflects a healthy state) or dysbiosis (when the microbiome affects the host negatively) [7]. Thus, gut microbiome represents one of the greatest, still undiscovered fields of possible markers and early predictors showing an association of bacteria with metabolic diseases, cancers, inflammatory responses and cognitive abilities. Microbiomes can be described at several levels, for instance, the taxonomic level resolution/granularity; from the phylum; via class, order, genus and species; or through their biological functions, such as producers of short-chain fatty acids (SCFAs) or bioactive molecules (like kynurenine pathway) or general pathogenic or opportunistic characteristics. So far, most taxonomic studies in humans have focused on microbiome analysis at the genera or phylum levels. However, it is now clear that some strains from within certain phyla are more beneficial than others, prompting researchers to zoom in on lower taxonomic levels and investigate microbial intricacies at the species level.

The taxonomic patterns clearly correspond with complex metabolic interactions between the host and their microbial communities; therefore, the interplay is more complicated than previously thought [8]. For instance, the bacterial species producing SCFAs (such as acetate, propionate and butyrate) from nutrients in the large intestine of humans thrive in the presence of prebiotics and dietary fibre [9]. Therefore, it is possible to stimulate microbial gut flora through specific diet and lifestyle and therefore prevent the development of dysbacteriosis.

### Gut Microbiome and Hyperglycaemia

The gut microbiome strongly affects glucose metabolism; in addition, the type of sugars and fats being consumed by the host also have an impact on the type of bacterial species the human gut is colonized with [10]. There is a reciprocal relationship between diet and gut microbiome composition, along with molecules produced by microflora that pass into the host’s circulation. The eubiotic gut epithelial barrier is maintained by a healthy, diverse microbiome composed primarily of four phyla: *Bacteriodetes*, *Firmicutes*, *Actinobacteria* and *Proteobacteria*. Phylogenetically related groups of bacteria, namely *Proteobacteria* and *Enterobacteriaceae*, were previously associated with poor glycemic control and with negative metabolic syndromes, including obesity, insulin resistance and impaired lipid profile [11,12,13]. The combined data from the past decade’s cohort studies strongly support the hypothesis of a prediabetes-associated microbial pattern that differs from healthy microflora [14,15,16]. Gathered here, research data comparing healthy versus diabetic gut microflora unequivocally confirm that there are statistically significant differences in microbiome between these groups. Butyrate producers are significantly reduced in prediabetes and type 2 diabetes, suggesting a clear metabolic deficit in these individuals [17,18]. Bacteria producing butyrate show strong anti-inflammatory properties, hence promoting healthy intestinal barrier [18]. Short-chain fatty acid (SCFA) producers break down non-absorbed carbohydrates and convert them into beneficial by-products used by host cells. Butyrate producers make up a functional group rather than taxonomic group and are either Gram-positive or Gram-negative [18]. SCFAs, especially butyrate, have been shown to improve ion absorption, intestinal barrier function, cell differentiation, motility and immune regulation at the intestinal level [19]. Additionally, SCFAs significantly improve electrolyte balance due to their non-ionic diffusion and SCFAs transporters. Since SCFAs are utilizing HCO3- exchange to enter the colonic cells, they contribute to the reduction in intracellular oxidative stress and stimulation of Sodium Chloride uptake [20]. Recent findings also confirm importance of the Tryptophan (Trp) metabolic pathway in multiple diseases inclusive of T2DM. Metabolites of Tryptophan such as kynurenine, kynurenate, xanthurenate and quinolinate and L-tryptophan increased in circulation in insulin-resistant individuals [21,22], whilst indole propionate was negatively associated with T2DM [22]. Indole-propionate-associated bacteria span three phyla, *Firmicutes*, *Actinobacteria* and *Bacteroidetes*, and are major SCFA producers, utilizing complex carbohydrates [22]. Therefore, when analyzing the microbiome, it is insufficient to only describe the microbiome at the phylum level, as there is a need to look into the lower taxonomic level to ensure more precise conclusions and higher specificity. The lower taxonomic information would allow the design of tailored microbial therapy to treat these metabolic inefficiencies.

The aim of our study was to gather already published information on which bacterial taxa were associated with development of T2DM. We have selected the articles on the basis of study design similarity, namely the same research question, the Illumina sequencing method and similar analytic protocols. Based on the collected information, we could not however determine if the recorded changes in microbiome occurred prior to diabetes or due to developing diabetes; therefore, we prompt the reader to interpret the association cautiously without linking it to the causation of T2DM.

## 2. Materials and Methods

A systematic review of the literature describing clinical cohort studies was performed on Ovid MEDLINE, Ovid Embase, PubMed and Google Scholar adhering to PRISMA guidelines [23]. Data were extracted from each published cohort study by a structured survey. The search terms used for the search included “gut microbiome AND T2DM AND Illumina” and “gut microbiome AND Type 2 Diabetes Mellitus AND Illumina”, with limits to human cohort studies published between 2010 and 2024. The information was indexed using controlled vocabulary and keywords (NVivo). Two reviewers screened each of the records, working independently. Exclusion criteria were applied to studies in which sequencing methods differed substantially and studies where only targeted quantitative PCR was undertaken. Further exclusion criteria were applied to the duodenal microbiome, with the limitation of the gut microbiome being sampled by stool analysis only. Studies evaluating treatment groups were included only if they had T2D untreated and non-diabetic control groups. Data were extracted from each published cohort study by structured survey.

We retrieved 10,932 original articles and review papers published during 2010–2024. We excluded articles in which 2TDM treatment was solely evaluated (Figure 1). Our study yielded 10,932 unique articles, of which 20 observational cohort studies were included in our analysis along with 1 machine learning association study. The presented cross-sectional studies collectively covered a population of 66,022 people, with 1800 in the 2TD group, 2904 in the prediabetic group and 61,318 in the control group (Appendix A shows all the details for each selected study). The selected studies covered a diverse population from various geographical locations and various ethnicities represented in Figure 2. Publications with missing or unclear methods of sequencing or omissions of primers used were excluded from the analysis. To assess certainty (or confidence) in the body of evidence for an outcome, we considered only reports where the *p*-value was significantly low.

### Risk of Bias Assessment

All studies were evaluated for methodological quality using the Joanna Briggs Institute (JBI) critical appraisal tool. The largest risk of bias in almost all studies was related to unassessed potential confounding, as most studies did not adjust for confounders, such as diet and antibiotics use, which could significantly influence the gut microbiome. Overall, the risk of bias for the studies was categorized as low or moderate (Appendix A). Since case–control studies are observational and retrospective in nature, they are prone to several types of bias, including recall bias, selection bias, confounding bias, temporal ambiguity and misclassification bias. In this study, we specified inclusion criteria very clearly to ensure the comparability of data generated by similar methods of analysis, therefore avoiding selection bias.

## 3. Results

Most of the studies (86%) analyzed here, covering population of different geographical locations (Figure 2) confirm that type 2 diabetes and prediabetes groups had significantly lower alpha diversity scores measured either by the Shannon index score or the Chao index. In general, all studies consistently reported that healthy microflora differed significantly from patients with T2D by certain species depletion and enrichment. The lower taxonomic analysis across all studies showed consistency in findings of type 2 diabetes being negatively correlated with bacteria, as shown in Figure 3A. Depletions of *Akkermansia* spp., *Ruminococcae*, *Faecalibacterium* spp., *Clostridiaceae*, *Bifidobacterium* spp. and *Bacteroides* spp. were all negatively correlated with T2DM across the highest number of cohorts included in this study. Consistent across all cohort studies, *Shigella*, *Escherichia* and *Ruminococci* were positively associated with the diabetic state, where increases in the abundances of those species were strongly linked to T2D and T2D disease progression (longitudinal cohort studies). Out of 20 cohorts, 7 reported increases in *Lactobacillae* and *Dorea*, and 1 study reported an increase in *Lachnospiracea* (to which *Dorea* belongs at a higher taxonomic level) [24,25,26,27]. Additionally, four cohorts reported increases in *Proteobacteria* phylum without lower taxonomic details (Figure 3B). Only 1 study out of the 20 analyzed here reported decreases in the abundances of *Blautia* and *Anaerostipes* in T2D participants, hence reporting those as positive species associated negatively with T2D progression [28,29,30]. Diener et al.’s study found that higher levels of *Blautia* and *Anaerostipes* were associated with lower areas under the glucose curve and normal beta cell function (FDR-adjusted LRT, *p* < 0.05); however, these results were not confirmed by the other studies analyzed here. Two studies that qualified under our selection criteria for cohort design and methodology reported results to the contrary: *Anaerostipes* and *Blautia* were reported as genera that increased abundance levels correlated positively with T2D [31,32]. The discrepancy might be due to use of metformin treatment in Diener et al.’s study, which is widely known to modify the microbiome. The increase in the abundance of bacteria from the genus *Butyricimonas* was also reported to counteract T2DM, which when zoomed in on the family level showed the prevalence of *Christensenellaceae* and *Rikenellaceae* [33]. All the cohort studies analyzed in this manuscript are consistent with the hypothesis of butyrate-producers’ abundance having a negative correlation T2D onset, suggesting that ethnical or geographic differences do not interfere with metagenomic markers. Metagenomic markers could be defined as specific genetic sequences or microbial signatures identified through metagenomic analysis that reflect the composition, diversity, or functional potential of a microbial community in a given environment. However, it has also been reported that the gut microbiome differs across geographic regions, with multiple species being good predictors for T2DM in some studies whilst showing no predictive value in other locations (e.g., China vs. Sweden [14,34]). Some studies, like the Finnish cross-populational study, zoomed in on the lowest taxonomic levels, revealing four to be consistently associated with T2D bacterial species: *Clostridium citroniae* (hazard ratio [HR] 1.21; 95% CI 1.04–1.42), *C*. *bolteae* (HR 1.20; 95% CI 1.04–1.39), *Tyzzerella nexilis* (HR 1.17; 95% CI 1.01–1.36) and *Ruminococcus gnavus* (HR 1.17; 95% CI 1.01–1.36) [31]. Moreover, the same study showed that at the higher taxonomic level, bacteria belonging to genera such as *Ruminococci*, *Blautia* and *Egghertella* are correlated with the incidence of T2D [28,30,35]. This review shows how scarce the whole genome sequencing data and the information on the microbiome at the species-level taxonomic resolution are, underlying the need for such studies to drive innovation in this field. A common discrepancy across studies presented includes *Faecalibacterium*, which in some studies analyzed here (3 out of 21) was positively associated with T2D, whilst in 6 other large cohorts, it was negatively associated with T2D and prediabetes [36]. This discrepancy in findings might be due to insufficient taxonomic analysis at the species level or geographic location of the study. For example *Faecalibacterium prausnitzii* is a Gram-negative spore former and a butyrate producer, the abundance of which has been negatively associated with inflammatory bowel disease [37,38,39]. In our analyses of 21 studies, we found bacteria reported as positive and negative in different studies (See Appendix A). Overall, there were 20 bacterial taxa reported as discrepant across studies.

## 4. Discussion

In the past decade, there were numerous cohort clinical observational studies aimed at revealing gut microbiome differences between diabetic and healthy subjects. However, just a few of these studies involved enough large population samples to draw general conclusions. Many studies of the gut microbiome showed that gut microbial flora might differ due to diet, geographical location, ethnic origin and many more confounding factors; therefore this systematic review takes a unique cross-sectional approach to gather the data from multi-ethnic, similarly designed studies. We have selected studies with a similar methodological design, namely a metagenome sequencing method using similar primers or the whole genome, to ensure comparability. The major limitation of our review method was that a small sample of studies adhered to the same microbiome analysis methodology. Also, association does not always imply causality. Therefore, we report the association between microbial communities and type 2 diabetes without inferring causality. The combined results represent a cohort of 65,754 people from a global population, covering all ethnicities and multiple geographic locations. Here, we confirmed specific microbial patterns of gut microbes that could indicate prediabetes or type 2 diabetes. The most recent metadata analysis by Gurung et al., 2020 summarized 42 human studies in which type 2 diabetes was investigated in terms of the microbiome. According to these results, genera negatively associated with T2D were reported as follows: *Bifidobacterium*, *Bacteroides*, *Faecalibacterium*, *Akkermansia* and *Roseburia* (increases in these species would be associated with lower T2D risk) [40,41,42,43,44]. Our results consider the most recent cohort studies over the past decade, showing that lower taxonomic classification is more informative and could help in selecting some taxa as predictive markers. Additionally, we would like to emphasize the need for a larger amplicon sequencing or whole genome sequencing approach to allow more precise readouts with the highest resolution for species. The results of our analysis confirm the protective capabilities of the taxon *Anaerostipes*, the members of which are butyrate producers. This is in line with previous findings that butyrate content in the lumen is associated with lower Peroxisome proliferator-activated receptor gamma activity (it is a protein that regulates genes involved in energy metabolism), increased glycolysis and lower oxygen consumption [45]. The analysis of these referenced cohort studies also confirms the positive effect of *Faecalibacterium*; however, it might be worth investigating what is the optimal abundance of this genera, as few other studies determined it to be a negative species. Other SCFAs producers from the *Clostridiacea* class were also confirmed in our study to be negatively correlated with type 2 diabetes. The higher abundance of this succinate-consuming species was earlier correlated with increased levels of propionate and butyrate production in the gut [46]. The causal link of diabetes and the decrease in the *Clostridiacea* class could be explained by the decrease in production of SCFAs, leading to a depletion of glucagon-like peptide 1 (GLP-1) and insulin production, since these SCFAs can lead to the secretion of GLP-1 by binding to G protein-coupled receptors (GPCRs) from L-cells; GLP-1 promotes insulin secretion and beta-cell proliferation [47,48]. However, one must be careful when interpreting butyrate producer deficiency in the microbiome as directly linked to diabetes, as the relationship with diet is very complex and many confounders are involved. It is widely known that various factors, such as diet, age and the use of medications like antibiotics, can lead to changes in the gut microbiota. Among these, diet has been identified as the most influential factor affecting the human gut microbiome. For instance, fermentable dietary fibers provide an energy source for beneficial gut bacteria, promoting the production of SCFAs and supporting microbial diversity. Therefore, these studies cannot conclusively prove the role of the gut microbiota in T2DM development, since they were all performed in already-diagnosed T2DM patients whose gut microbiota could also have been altered by several of these confounding factors. Nevertheless, up until now, the research has shown that the abundance of butyrate-producing bacteria differs between healthy individuals and those with diabetes, suggesting that the gut microbiome may play a role in regulating blood glucose levels. Our study confirmed the consistent association of some bacteria with diabetes without implying the causation of diabetes. Notably, diet has been identified as the primary factor exerting the most substantial impact on the human gut microbiota; for this reason, one may deduce that the development of diabetes might start with a diet that triggers changes in the microbiome [41].

An important multi-omic study by Zhao et al., 2017 investigated metabolic profiles using Liquid Chromatography/Mass Spectrometry and Gas Chromatography/Mass Spectrometry methods for fecal metabolome, revealing that the concentrations of SCFAs were predominantly reduced in the T2D group (Kruskal–Wallis H-test, *p* < 0.01, BMI- and age-adjusted *p* < 0.01) [49]. In addition, the same study found levels of lysophosphatidylcholine (LPC), cholic acid and palmitoyl carnitine to be higher in the T2D group by a 4.51- to 13.84-fold change (Kruskal–Wallis H test, *p* < 0.001). LPC is known for its demyelination properties in vivo, but it also could be found in circulation at various levels as it is essential component of normal human brain development.

Interestingly, in line with taxonomic analysis, another gut microbiome metabolomics study revealed that bacterial genes highly associated with T2D encoded tyrosine degradation enzymes, pentose phosphate pathway proteins, lactose, galactose and butyrate production proteins [30]. These findings support the hypothesis of a specific bacterial composition that might distinguish a healthy gut microbiome from a T2D one, further allowing the analysis of diet and its effect on microflora and metabolic by-products. Steps towards the improvement of gut microbiome have already been taken via probiotic and prebiotic therapy; however, there are multiple novel strains discovered in the past decade, with proven efficiency in vitro mouse models, awaiting clinical trials. In particular, the administration of *Akkermanisa muciniphila* has been confirmed by several studies as a microbial probiotic able to improve glucose tolerance and insulin resistance in mice [50,51]. In all the cohort studies under analysis for the purpose of this review, *Akkermansia muciniphila* proved to be consistently increased. *A. muciniphila* was recently discovered to be of major importance in gut health, contributing to the reduction in obesity and decrease in metabolic disorders in diabetic mouse models [52]. *A. muciniphila* can restore the mucosal layer in the gut and is considered to reduce gut permeability, hence acting as anti-inflammatory probiotic species [53]. Overall, consistently in all studies published so far, the abundance of *Akkermansia* spp. has been inversely correlated with diabetes, Body Mass Index (BMI), inflammatory bowel disease (IBD) and autism [54,55,56]. The bacteria are also commercially available on the market in the UK and Europe, suggesting their large therapeutic potential in the field of microbial sciences. Our study confirms that the gut microbiota represent an important modifiable factor to consider when developing precision medicine approaches for the prevention and/or delay of T2D. The ideal future work to continue this study would be a stratified meta-analysis, gathering the raw sequencing data from the studies reviewed here, which could potentially reveal more information and allow more sensitive statistical evaluation. For that reason, we strongly support repositories with open access policies such as the European Genome-Phenome Archive [57] or the NIH Human Microbiome Project, which would allow further discoveries in this exciting field.

Throughout this study, we also recorded bacterial taxa that were reported inconsistently across different studies (see Appendix A). For example, *Paraprevotella* was found to be positively associated with T2D in two independent studies with a combined sample size of 579, while a larger study by Wang et al. (n = 2895) reported a negative correlation with the disease [14,24,42]. These discrepancies could have been due to confounding factors such as differences in study design, demographics, diet or geographic location/ethnicity. A typical discrepancy spotted in our study, related to the effect size, was a single cohort by Ahmad et.al reporting *Prevotella* species as being negatively associated with diabetes against four large cohort studies reporting these species as being positively associated with disease.

The primary limitation of all studies included in this analysis is their cross-sectional design, which introduces selection bias. Although the inclusion and exclusion criteria were clearly defined in each study, the lack of longitudinal data limits the ability to assess the true predictive value of the identified bacterial taxa. These findings highlight the need for further proof-of-principle studies, particularly longitudinal cohort studies that could stratify participants based on common microbial patterns. Such a design would enable the prospective evaluation of diabetes incidence rates.

Another important challenge in studying the association of the gut microbiome with type 2 diabetes is the possible influence of medications, such as metformin. In our study, we selected publications of cohorts that did not aim at metformin treatment evaluation; however, in a few studies, metformin groups were specified as additional groups of comparison (controlled factor). This was an important part of inclusion criteria, as it has been demonstrated that studies comparing treatment-naive individuals with diabetes to those treated with metformin have shown that many associations previously thought to be linked to disease progression were actually a result of the treatment and were not present in individuals without a history of metformin use. It is now proven that changes in lifestyle or diet can influence the gut flora independently of disease, so incorporating these multifactorial dependencies into future diabetes prediction models could help to improve our understanding of the epidemiological causes.

## Figures and Tables

**Figure 1 microorganisms-13-00134-f001:**
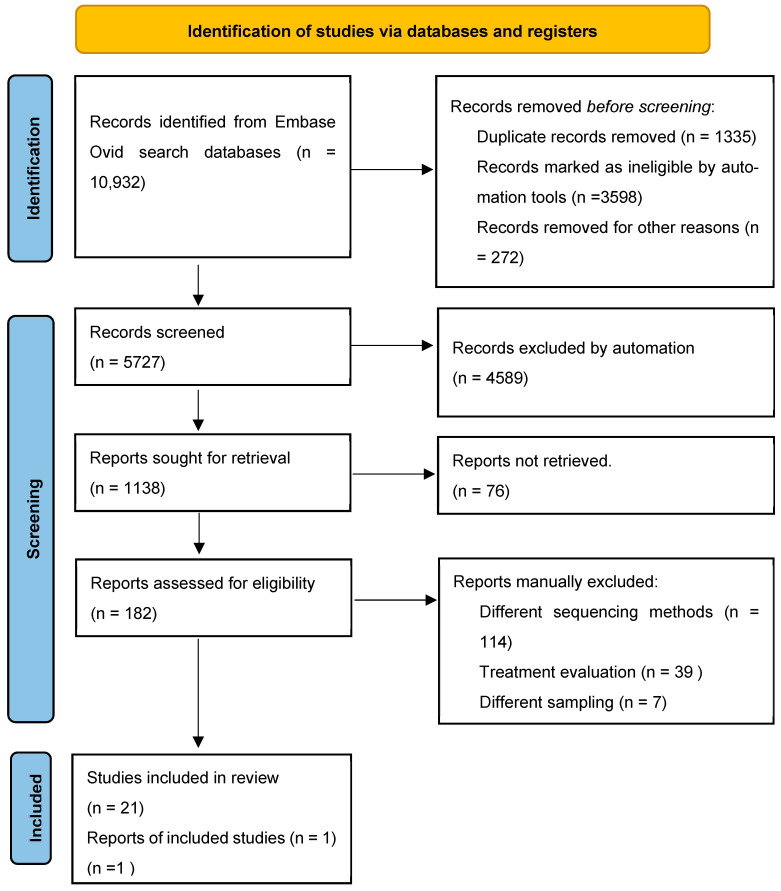
PRISMA diagram of study identification and selection.

**Figure 2 microorganisms-13-00134-f002:**
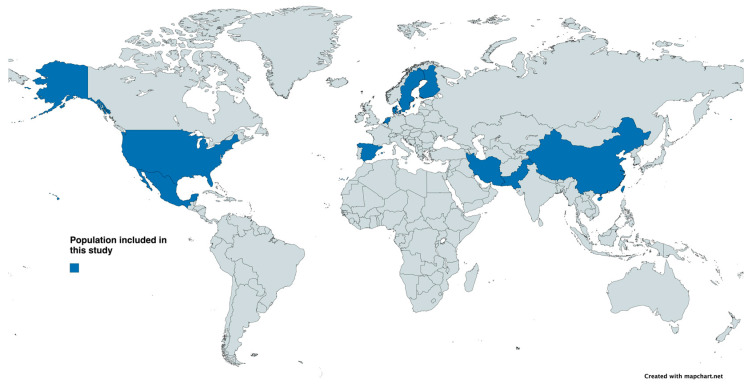
Map illustrating the geographical distribution of the populations included in this study.

**Figure 3 microorganisms-13-00134-f003:**
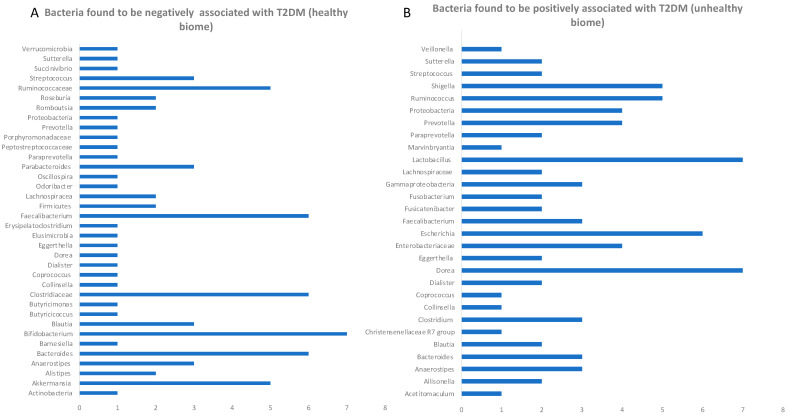
A cumulative list of bacterial taxa reported in the reviewed publications. (**A**) Number of studies describing bacterial taxa negatively associated with type 2 diabetes mellitus. (**B**) Number of studies in which specific bacterial taxa were found to be negatively associated with type 2 diabetes.

## Data Availability

The data could be accessible upon request.

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
