# Peer review of "Understanding Patterns of the Gut Microbiome May Contribute to the Early Detection and Prevention of Type 2 Diabetes Mellitus: A Systematic Review"

_microorganisms, 2025, doi:10.3390/microorganisms13010134_

Round 1
Reviewer 1 Report
Comments and Suggestions for Authors
The article successfully synthesizes 21 studies from various regions and ethnic groups, yielding results with considerable representativeness. It focuses on the relationship between gut microbiota and type 2 diabetes mellitus (T2DM), a field with significant scientific and clinical implications. However, I have the following concerns:
1. The study covers diverse geographical populations but lacks a clear analysis of regional differences in microbiome composition and T2DM association.
2. The paper effectively synthesizes results but could benefit from additional quantitative meta-analyses to statistically assess the strength of reported associations.
3. Figures are informative but require better labeling and captions to ensure accessibility for readers unfamiliar with specific microbiome terms. The “Faecalibacterium,” in 2A should be “Faecalibacterium”. In Figure 2, it shows (a) and (b) in the figure legend but shows A and B in the figure.
4. The supplementary table is mentioned but not adequately integrated into the main text. Consider referencing it more explicitly in relevant sections.
5. The reported lower alpha diversity in T2DM and prediabetes groups is consistent but could be strengthened by discussing potential confounding factors like diet and medication use. The potential confounding effect of medications such as metformin on microbiome composition is not deeply discussed. This could be addressed further.
6. While the discussion identifies key microbial taxa, it could delve deeper into molecular pathways (e.g., SCFA production) linking microbiota and T2DM progression.
7. Repeated use of complex terms like “short-chain fatty acid producers” could be simplified or explained for improved readability.
8. Discrepancies in taxa associations across studies (e.g., Faecalibacterium) are noted but require more detailed exploration of possible causes, such as methodological variations.
9. Some technical terms (e.g., "metagenomic markers") are not defined for non-specialist readers and should be clarified.
10. Some included studies have small sample sizes, which may reduce statistical power. This limitation should be discussed more explicitly.
Comments on the Quality of English LanguageThe manuscript is generally well-written but contains minor grammatical errors and instances of awkward phrasing (e.g., "derby prevent development").
Author Response
REVIEWER 1
The article successfully synthesizes 21 studies from various regions and ethnic groups, yielding results with considerable representativeness. It focuses on the relationship between gut microbiota and type 2 diabetes mellitus (T2DM), a field with significant scientific and clinical implications. However, I have the following concerns:
- The study covers diverse geographical populations but lacks a clear analysis of regional differences in microbiome composition and T2DM association.
Response 1: We would like to thank the reviewer for this well-noticed fact of the study being cross-sectional. We would like to point the reader to Supplement table 1 where all the studies are listed with regional information. To address the diversity, we have created a Figure 2.in the main manuscript body, showing the geographical locations of the studies analysed here.
- The paper effectively synthesizes results but could benefit from additional quantitative meta-analyses to statistically assess the strength of reported associations.
Response 2: Excellent suggestion. Here, we aimed at systematic review and not a meta-analyses as not all data is easily obtainable in the raw format given time-constraints. However, we would like to continue this research by contacting the authors of considered studies with request for their sequencing data and re-analysing all the data independently.
- Figures are informative but require better labeling and captions to ensure accessibility for readers unfamiliar with specific microbiome terms. The “Faecalibacterium,” in 2A should be “Faecalibacterium”. In Figure 2, it shows (a) and (b) in the figure legend but shows A and B in the figure.
Response 3: We appreciate the reviewer’s valuable feedback. In response, we have corrected the typos in Figure 2 and updated the figure labels to align with the caption descriptions. Additionally, due to the inclusion of a new figure, Figure 2 has been renumbered to Figure 3. While the publication is primarily aimed at a microbiology-savvy audience, we believe the captions provide sufficient clarity on microbial terms, which can be easily looked up using online resources if needed.
- The supplementary table is mentioned but not adequately integrated into the main text. Consider referencing it more explicitly in relevant sections.
Response 4: Very good suggestion. We have updated Supplementary data with additional tables and referenced the data within the text where appropriate (highlighted in red).
- The reported lower alpha diversity in T2DM and prediabetes groups is consistent but could be strengthened by discussing potential confounding factors like diet and medication use. The potential confounding effect of medications such as metformin on microbiome composition is not deeply discussed. This could be addressed further.
Response 5: The potential confounders are now included in discussion as well as in results section under a paragraph entitled ‘’Risk of bias’’. These studies which evaluated metformin use in treatment of diabetes were excluded, as per our exclusion criterion mentioned in materials and methods. However some cohorts included in the study also compared 3rd group of treatment (we have highlighted this in Supplementary table 1) hence 3 studies in which metformin was used as additional group were included in our study. We have made a discussion point about metformin in the discussion section (line 363-374).
- While the discussion identifies key microbial taxa, it could delve deeper into molecular pathways (e.g., SCFA production) linking microbiota and T2DM progression.
Response 6: We would like to thank the reviewer for pointing the importance of these pathways in study of microbiome, however due to vastness of resources on this topic we had decided not to expand on this further. We have made a significant introduction of SCFAs in line 44-60 ad 85 to 100. Then we have also discussed the relevant pathways on page 7, line 274 to 307. Additionally we have extensively described other pathways such as tyrosine-degradation, pentose-phosphate pathway on page 8 , line 317-344. We agree that these pathways require separate focus in another review.
- Repeated use of complex terms like “short-chain fatty acid producers” could be simplified or explained for improved readability.
Response 7: We would like to thank the reviewer for picking this up. We agree with the comment, and accordingly changed the ‘’short-chain fatty acid’’ term into SCFAs abbreviation introduced at the beginning of the manuscript.
- Discrepancies in taxa associations across studies (e.g., Faecalibacterium) are noted but require more detailed exploration of possible causes, such as methodological variations.
Response 8: Whilst we appreciate this comment, we would like to emphasize that the aim of our study was to eliminate methodological variations by selecting the publications which followed the same analytical protocols. Therefore, the methodological variations, variations in measurements of exposure, subject inclusion and exclusion criteria were all reduced. However, we have mentioned other biases in both discussion section and materials and methods section.
- Some technical terms (e.g., "metagenomic markers") are not defined for non-specialist readers and should be clarified.
Response 9: We are grateful for the reviewer’s comment. We have included a sentence explaining the meaning of this complex term. The text is highlighted in red, on page 5, line 219-222.
- Some included studies have small sample sizes, which may reduce statistical power. This limitation should be discussed more explicitly.
Response 10: We have listed all the studies in the Supplementary materials, where sample size per each study is specified (Supplementary Table 1). Additionally, for the bacterial species/taxa which we found discrepantly recorded by different studies we have added a separate table (Supplement table 5) with author’s discussion on why the discrepancy might have occurred. Additionally, we have included a paragraph to the discussion section (Line 343-359, highlighted in red) where the small sample size is defined as study design bias. Overall, the aim of this review was to collect small studies and quantify them to see if similar results were observed to compare with larger studies. Ideally the next step would be to conduct a meta-analysis on all data.
Reviewer 2 Report
Comments and Suggestions for Authors
In the present review, the authors focus on Type 2 Diabetes Mellitus (T2DM) and its increasing prevalence in developing countries during the last years.
In this vein, they selected studies based on similar methodological approaches in order to limit heterogeneity and bias. The review collects large data from 65,754 individuals. Yet, there is a discussion on the role of the microbiome in relation to diabetes and the possibility of its modulation by diet ,probiotics and other.Also, they recommend development of standard operating protocols ,such as STORMS.
However, it is necessary to specify the criteria for selection of designs and methodologies. Yet, here is no in-depth discussion of the specific biases (e.g., publication bias, selection bias.Statistical methods used are not explained.
In general it is a well written paper with nice representations and tables and could be published after clarification of the above.
Author Response
We would like to thank the reviewer for their valuable insights and suggestions. We have now included bias assessment data performed by 2 independent researchers in the Supplementary data. All studies were evaluated for methodological quality using the Joanna Briggs Institute (JBI) critical appraisal tool. We have included the information in materials and methods section highlighted in red, page 3 line 137. Because current studies have a high methodological heterogeneity and risk of bias, we have selected studies which adhered to similar design and methodology.
We have also included comments on confounders and biases in the discussion section to ensure the reader is aware of all possible biases affecting this study and studies under analysis.
Round 2
Reviewer 1 Report
Comments and Suggestions for Authors
No more comments and suggestions.